Identification and characterization of Gypsophila paniculata color morphs in Sleeping Bear Dunes National Lakeshore, MI, USA

http://orcid.org/0000-0002-9898-2094 Yang Marisa L. 1
Rice Emma 2
Leimbach-Maus Hailee 2
Partridge Charlyn G. 2 partridc@gvsu.edu
1 Environmental Science Policy and Management, University of California, Berkeley , Berkeley, CA , USA
2 Annis Water Resources Institute, Grand Valley State University , Muskegon, MI , USA
Culham Alastair
Electronic publication date: 2019 Jun 17
Publication date: 2019
Volume: 7
Electronic Location ID: e7100
Received 2018 Mar 14; Accepted 2019 May 8
Copyright: © 2019 Yang et al.
Copyright year: 2019
Copyright holder: Yang et al.
License: This is an open access article distributed under the terms of the Creative Commons Attribution License, which permits unrestricted use, distribution, reproduction and adaptation in any medium and for any purpose provided that it is properly attributed. For attribution, the original author(s), title, publication source (PeerJ) and either DOI or URL of the article must be cited.
License URL: https://creativecommons.org/licenses/by/4.0/

Keywords: Gypsophila paniculata, Invasive species, Color morphs

Funding: Environmental Protection Agency—Great Lakes Restoration Grant GL-00E01934 National Science Foundation Research Experience for Undergraduates Grant 1461249 This work was supported by an Environmental Protection Agency—Great Lakes Restoration Grant (No. GL-00E01934) and a National Science Foundation Research Experience for Undergraduates Grant (No. 1461249). The funders had no role in study design, data collection and analysis, decision to publish, or preparation of the manuscript.

==============================
Background

Gypsophila paniculata (baby’s breath) is an invasive species found throughout much of the northwest United States and western Canada. Recently, plants exhibiting a different color morphology were identified within the coastal dunes along eastern Lake Michigan. The common baby’s breath (G. paniculata) typically produces stems that are purple in color (purple morph), while the atypical morph has stems that are green-yellow (green-yellow morph). The purpose of this study was to characterize these newly identified morphs and determine if they are genetically distinct species from the common baby’s breath in order to assess whether alternative management strategies should be employed to control these populations.

Methods

We sequenced two chloroplast regions, ribulose-bisphosphate carboxylase gene (rbcL), and maturase K (matK), and one nuclear region, internal transcribed spacer 2 (ITS2), from the purple morphs and green-yellow morphs collected from Sleeping Bear Dunes National Lakeshore, MI, USA (SBDNL). Sequences were aligned to reference sequences from other Gypsophila species obtained from the Barcode of Life Database and GenBank databases. We also collected seeds from wild purple morph and wild green-yellow morph plants in SBDNL. We grew the seeds in a common garden setting and characterized the proportion of green-yellow individuals produced from the two color morphs after 5-months of growth.

Results

Phylogenetic analyses based upon rbcL, matK, and ITS2 regions suggest that the two color morphs are not distinct species and they both belong to G. paniculata. Seeds collected from wild green-yellow morphs produced a significantly higher proportion of green-yellow individuals compared to the number produced by seeds collected from wild purple morphs. However, seeds collected from both color morphs produced more purple morphs than green-yellow morphs.

Discussion

Based upon these results, we propose that the two color morphs are variants of G. paniculata. Given the significant difference in the number of green-yellow morphs produced from the seeds of each morph type, we also suggest that this color difference has some genetic basis. We propose that current management continue to treat the two color morphs in a similar manner in terms of removal to prevent the further spread of this species.

Introduction

The Great Lakes sand dunes comprise the most extensive freshwater dune complex in the world, stretching over 1,000 km2 in Michigan alone. Within northwest Michigan, the sand dunes ecosystem is vital both environmentally and economically. It is home to a number of threatened and endangered species, including piping plover (Charadrius melodus) and Pitcher’s thistle (Cirsium pitcheri). Colonization of invasive species in this region has the potential to significantly alter the biological composition of these native communities (Leege & Murphy, 2001; Emery et al., 2013). One invasive species of significant concern is the perennial baby’s breath (Gypsophila paniculata). In 2015, baby’s breath was listed by the Michigan Department of Natural Resources as a “priority” invasive species for detection and control in Michigan’s northern lower peninsula (Department of Natural Resources (DNR), 2015). Since its colonization in the region it has spread along a 260 km stretch of the Michigan shoreline. Baby’s breath produces a large taproot system that can extend down to four meters in depth, which likely helps it outcompete native vegetation for limited resources (Darwent & Coupland, 1966; Karamanski, 2000). In addition, while many of the vulnerable and endangered plant species in these areas are seed limited, (e.g., Pitcher’s thistle produces approximately 50–300 seeds per plant total or “per lifetime” (Bevill, Louda & Stanforth, 1999)), baby’s breath can produce up to 14,000 seeds per plant annually (Stevens, 1957), effectively outcompeting native species in terms of overall yield. This has led to baby’s breath composing approximately 50–80% of the ground cover in some areas (Karamanski, 2000; Emery et al., 2013).

One concern with current management efforts is that anecdotal evidence suggests there may be a new baby’s breath variant within the Michigan dune system. In 2011 and 2012 The Nature Conservancy (TNC) removal crews reported baby’s breath plants with different character traits than what is commonly observed (The Nature Conservancy (TNC), 2014). The atypical morph has stems and leaves that are lighter in color and more yellow than the common G. paniculata purple morph (Figs. 1A–1C). The purple morph has a thick taproot (four to seven cm in diameter) just below the caudex that remains unbranched for approximately 60–100 cm (Darwent & Coupland, 1966). Severing just below the intersection of the caudex and the taproot is where manual removal efforts target to limit regrowth. However, TNC removal crews suggested that the atypical green-yellow morph’s root system seemed to be more diffuse, making it harder to identify a primary taproot and thus, harder to sever without the potential for regrowth (The Nature Conservancy (TNC), 2014). Currently, these green-yellow morphs are treated with herbicide application (glyphosate) when observed; however, if this is a newly invaded baby’s breath species and it continues to spread into areas where threatened or endangered species are present, removal methods will be a primary concern and alternative management strategies may need to be considered for these populations.

Figure 1 (A) Green-yellow morph, (B) common purple morph, and (C) stem of the green-yellow and purple baby’s breath morph found in Sleeping Bear Dunes National Lakeshore.

One of the first steps toward adapting current management strategies for this invasive is to identify whether the green-yellow morph is a genetically distinct species from the purple morph. While G. paniculata is the dominant invasive baby’s breath species in northwest Michigan, a number of other species have been introduced to North America and, specifically, the Great Lakes region (Pringle, 1976; Voss & Reznicek, 2012). For example, G. elegans, G. scorzonerifolia, G. muralis, and G. acutifolia have been collected within Michigan (Pringle, 1976; Reznicek, Voss & Walters, 2011; Voss & Reznicek, 2012), and G. perfoliata is reported to have become naturalized in the United States (Pringle, 1976). G. muralis is an annual and has a very distinct morphology compared to the other Gypsophila species identified around the Great Lakes. It typically only reaches 5–20 cm in height, has linear leaves, and commonly produces white to pink flowers (Barkoudah, 1962). G. elegans, also an annual, is commonly sold in this region in commercial wildflower packets. It typically has a smaller taproot compared to G. paniculata, and its coloration can be similar to that observed for the green-yellow morph. G. scorzonerifolia and G. acutifolia are perennials and specimens of these species have been collected in counties within the Great Lakes dune system that also contain G. paniculata infestations (Voss, 1957; Pringle, 1976). Both G. scorzonerifolia and G. acutifolia have a deep taproot and are similar in height to G. paniculata, and thus, can superficially resemble G. paniculata (Voss, 1957; Pringle, 1976). However, both can be distinguished from G. paniculata in that their leaves tend to be longer and wider, and the pedicels and calyces are glandular as opposed to glabrous in G. paniculata (Voss, 1957; Pringle, 1976). Given the potential for these other species to invade the fragile habitat of the Michigan dune system, the goal of this work was to characterize the genetic relationship between the newly recognized green-yellow morph and the common purple morph to determine if they are the same species.

Methods and Materials

DNA extraction, amplification, and sequencing

We collected leaf tissue from one green-yellow morph and 16 purple morphs in 2016 and an additional 15 green-yellow morphs in 2017 from Sleeping Bear Dunes National Lakeshore (SBDNL), Empire, MI, USA (specifically: 44.884941N, 86.062111W and 44.875302N, 86.056821W). Plant tissue collections were approved by the National Parks Service (permit ID SLBE-2015-SCI-0013). Leaf tissue was dried in silica gel until DNA extractions could take place. DNA was extracted using a Qiagen DNeasy Plant Mini Kit (Qiagen, Hilden, Germany). After extraction, the DNA samples were placed through Zymo OneStep PCR inhibitor removal columns (Zymo, Irvine, CA, USA) to remove any secondary metabolites that might inhibit PCR amplification. The DNA for each sample was then quantified using a NanoDrop 2000 (Thermo Fisher, Waltham, MA, USA).

The DNA of green-yellow morphs and purple morphs was amplified at three genetic regions: large subunit of the ribulose-bisphosphate carboxylase gene (rbcL), maturase K (matK), and internal transcribed spacer 2 (ITS2). A combination of ITS region and matK have been used to differentiate between other Gypsophila species (specifically, G. elegans and G. repens) in previous studies (Fior et al., 2006). The rbcL region was amplified using rbcL 1F and rbcL 724R primers (Fay, Swensen & Chase, 1997), matK was amplified using matK 390F and matK 1440R primers (Fior et al., 2006), and the ITS2 region was amplified using ITS2 S2F and ITS2 S3R primers (Chen et al., 2010). PCR reactions for all loci consisted of 1× Taq Buffer, 2.0 mM MgCl2, 0.3 μM dNTP, 0.08 mg/mL BSA, 0.4 μM forward primer, 0.4 μM reverse primer, and 0.5 units of Taq polymerase in a 20 μL reaction volume. The thermal cycle protocols consisted of the following: for rbcL, an initial denaturing step of 95 °C for 2 min, followed by 35 cycles of 94 °C for 1 min, 55 °C for 30 s, and 72 °C for 1 min. A final elongation step was performed at 72 °C for 7 min. For matK, the thermal profile consisted of 26 cycles of 94 °C for 1 min, 48 °C for 30 s, and 72 °C for 1 min, followed by a final elongation step at 72 °C for 7 min. For ITS2, an initial denaturing step of 95 °C for 2 min was applied, followed by 35 cycles of 95 °C for 30 s, 50 °C for 30 s, 72 °C for 1.5 min, and a final elongation step of 72 °C for 8 min. Successful amplification was checked by running the PCR product on a 2% agarose gel stained with ethidium bromide. PCR reactions were then cleaned using ExoSAP-IT PCR Product Cleanup Reagent (ThermoFisher, Waltham, MA, USA). Sequencing reactions were performed with the forward and reverse primers for each of the three regions. Sequencing reactions were cleaned using a Sephadex column (GE Healthcare Life Science, Marlborough, MA, USA) and sequenced on an ABI Genetic BioAnalyzer 3130xl (Applied Biosystems, Foster City, CA, USA). Out of the 16 green-yellow morphs a total of 13 were successfully sequenced for rbcL, 13 were successfully sequenced for matK, and 14 were successfully sequenced for ITS2. For the purple morphs a total of 15, 12, and 15 individuals were successfully sequenced for rbcL, matK, and ITS2, respectively.

Reference sequences for rbcL, matK, and ITS2 of other Gypsophila spp. were downloaded from either the Barcode of Life Database (BOLD) (http://www.barcodeoflife.org) or GenBank (https://www.ncbi.nlm.nih.gov/genbank/). We primarily focused on Gypsophila species with reported occurrences within the United States, but also incorporated other species if their information was available on BOLD. Sequences of the three regions were not always available for the same species, thus for rbcL these included G. paniculata, G. elegans, G. fastigiata, G. scorzonerifolia, G. perfoliata, and G. muralis. For matK the species included G. paniculata, G. elegans, G. fastigiata, G. scorzonerifolia, G. perfoliata, G. muralis, G. altissima, and G. repens. For the IST2 region, the species included G. paniculata, G. elegans, G. scorzonerifolia, G. perfoliata, G. repens, and G. acutifolia. Because sequences for all three regions were not available for all species, our merged phylogeny only contained G. paniculata, G. elegans, G. scorzonerifolia, and G. perfoliata reference sequences. The accession numbers and sequences for all reference species are provided in Table S1. All FASTA files corresponding to these data have been deposited in the Dryad database and sequences have been submitted to GenBank (ITS2: MG385003–MG385031, matK: MG603322–MG603346, rbcL: MG547346–MG547373).

Alignment and phylogenetic analysis

All successful sequences from our field samples, as well as sequences for other Gyposphila species obtained from BOLD or GenBank, were imported into the program MEGA7 (version 7.0.14) (Kumar, Stecher & Tamura, 2016) and sequences for each of the three regions were aligned both individually and with all sequences combined using Muscle (Edgar, 2004). The total number of base pairs (bps) aligned and analyzed for each region included: 427 bp for rbcL, 702 bp for matK, 201 bp for ITS2, and 1,330 bp for the three regions combined. All alignment parameters were kept at their default settings. Once aligned, we used MEGA7 to identify the most appropriate substitution model (rbcL: Jukes-Cantor, matK: Tamura 3-parameter, ITS2: Jukes-Cantor, all genes combined: Tamura 3-parameter with gamma distribution). We then created phylogenetic trees using a maximum-likelihood approach with 500-replicated bootstrap analyses, as well as using neighbor joining, and parsimony models. We also constructed a TCS haplotype network (Clement et al., 2002) based upon the combined sequences using the statistical parsimony approach (Templeton, Crandall & Sing, 1992) in the program PopART (v 1.7, http://popart.otago.ac.nz/).

Color morph germination

On May 8, 2018 we planted a total of 207 seeds collected from mature purple morphs and 255 seeds collected from mature green-yellow morphs from SBDNL. For the purple morph seeds, these were collected from a total of 14 plants that were sampled in 2016 (average 15 seeds per plant) and seven plants that were sampled in 2017 (average 1.7 seeds per plant). For the green-yellow morphs, these seeds were collected from a total of 17 plants (15 seeds per plant) in 2017. Plants were grown in the GVSU—Allendale greenhouse from May until August, 2018. The greenhouse was on a 17 h light/7 h dark cycle. The average day temperature was 21 °C and night temperature was 15 °C. In August the plants were transported to the greenhouse at AWRI-GVSU where they were allowed to grow until October 15, 2018. The greenhouse at AWRI-GVSU has no external lighting source or temperature controls, and thus more closely resembled seasonal day/night and temperature cycles. Plants were sampled after a decrease in temperature occurred (from a high of 23 °C on October 10, 2018 to a high of 11 °C on October 15, 2018). Previous greenhouse observations have found that the differences between the purple and green-yellow morphs can be best detected after a sudden drop in temperature (C.G. Partridge, 2017, personal observation). On October 15, we characterized the color of all individuals that successfully germinated and survived over the 5-month period. We used a chi-square analysis in the R statistical package (R Core Team, 2018) to determine if germination success differed between seeds from the two color morphs and whether the proportion of seeds that developed into green-yellow morphs significantly differed between seeds collected from mature purple and mature green-yellow plants.

Results

Our results indicate that the green-yellow morph identified in SBDNL is not a genetically distinct species from the common purple found throughout SBDNL. The rbcL, matK, ITS2, and combined dataset showed similar patterns with both the green-yellow morphs and the purple morphs clustering together. The phylogenies constructed from rbcL and matK independently show that the two color morphs cluster separately from G. fastigata, G. elegans, G. muralis, and G. repens. For the rbcL locus, the relationship of the color morphs to G. paniculata and G. scorzonerifolia was not resolved. Additionally, when we only examined the matK gene, the color morphs clustered separately but within a clade that also included G. altissima, G. scorzonerifolia, and G. paniculata. The ITS2 region was able to provide more resolution between G. paniculata, G. scorzonerifolia, G. acutifolia, and the color morphs, with the color morphs clustering with G. paniculata (Figs. 2–4; Figs. S1–S8) and separately from the G. scorzonerifolia and G. acutifolia clade. The same pattern was observed when all regions were analyzed together (Fig. 5), with the exception that this phylogeny did not include G. acutifolia. In addition, the TCS haplotype network shows that the purple and green-yellow morphs have shared haplotypes. These two haplotypes are only one mutation away from one another and the G. paniculata reference, while the next closest species, G. scorzonerifolia is 15 mutations away (Fig. 6). This further suggests that both color morphs are G. paniculata.

Figure 2 Phylogenetic analysis of the purple and green-yellow baby’s breath color morphs in relationship to other Gypsophila species based on the rbcL region.

The evolutionary history was inferred using maximum likelihood methods. For this analysis, we used a Jukes Cantor (JC) model of molecular evolution (Jukes & Cantor, 1969).

Figure 3 Phylogenetic analysis of the purple and green-yellow baby’s breath color morphs in relationship to other Gypsophila species based on the matK region.

The evolutionary history was inferred using maximum likelihood methods. For the analysis, we used a Tamura 3-parameter (T92) model of molecular evolution with uniform distribution (Tamura, 1992).

Figure 4 Phylogenetic analysis of the purple and green-yellow baby’s breath color morphs in relationship to other Gypsophila species based on the ITS2 region.

The evolutionary history was inferred using maximum likelihood methods. For the analysis we used a Jukes Cantor (JC) model of molecular evolution (Jukes & Cantor, 1969).

Figure 5 Phylogenetic analysis of the purple and green-yellow baby’s breath color morphs in relationship to other Gypsophila species based on the rbcL, matK, and ITS2 regions combined.

The evolutionary history was inferred using maximum likelihood methods. For the analysis, we used a Tamura 3-parameter (T92) model of molecular evolution with gamma distribution (Tamura, 1992).

Figure 6 A TCS haplotype network based on rbcL, matK, and ITS2 combined for the purple and green-yellow baby’s breath color morphs and the G. paniculata, G. elegans, G. perfoliata, and G. scorzonerifolia reference sequences.

The size of the ovals correspond to the haplotype frequency. The hash marks represent the number of mutations between each haplotype.

Of the regions analyzed for the green-yellow morphs, purple morphs, and reference sequences, rbcL was the most conserved sequence with an overall mean genetic distance (d) = 0.004, followed by matK (d = 0.015), and ITS2 (d = 0.038). For the ITS2 region, there were six purple morphs and one green-yellow morph that clustered together inside the G. paniculata branch (Figs. 4 and 5). Further examination of the electropherograms for these individuals show that they are likely heterozygous at position 138 of our aligned sequence and amplification bias of the “A” single nucleotide polymorphism (SNP) over the allele containing the “G” SNP is driving this pattern.

Color morph germination

Out of the 207 seeds that were collected from mature purple morphs and planted in the greenhouse, 82 successfully germinated and survived over the 5 month period (39.6%). Out of these 82 plants, only one green-yellow morph was produced (1.2%), while the remaining seeds all produced purple morphs. Out of the 255 seeds collected from mature green-yellow morphs and planted in the greenhouse, 105 successfully germinated, and survived over the 5 month period (41.2%). This was not significantly different than the proportion of plants that successfully germinated from the purple morph seeds (χ2 = 0.06, df = 1, p = 0.81). Of the 105 successfully germinated seeds from the green-yellow morph plants, 12 developed into green-yellow morph plants (11.4%), 91 developed into purple morphs plants (86.7%), and two plants could not be determined (they appeared to be green-yellow morphs but displayed some dark spots on the stem). The proportion of seeds that produced green-yellow individuals significantly differed between seeds collected from mature green-yellow morphs and seeds collected from mature purple morphs (χ2 = 5.9, df = 1, p = 0.015).

Discussion

Overall, our data suggest that the green-yellow morph is not a genetically distinct species from the purple morph, and that both morphs are G. paniculata. For all molecular markers used, the green-yellow and the purple color morphs grouped together. RbcL, matK, and ITS2 are common “barcode” regions used to delineate plant species (Newmaster, Fazekas & Ragupathy, 2006; Group et al., 2009; Chen et al., 2010; Yao et al., 2010; Stoeckle et al., 2011) and when used in combination they provided adequate resolution to separate out the Gypsophila species included in this study. In our data set, rbcL, and matK worked well to separate our color morphs from G. elegans, G. muralis and both of these species have been reported to occur in the Great Lakes region (Reznicek, Voss & Walters, 2011; Voss & Reznicek, 2012). While the morphology of G. muralis is very distinct from the color morphs in SBDNL, it was initially thought that G. elegans shared some similar traits to that originally described by the TNC removal crews and was a potential candidate species for the green-yellow color morph. Based upon these results, this is clearly not the case.

The phylogeny based on ITS2 region and the combined sequences provided the best resolution for assigning the relationship of our Gypsophila species. Like the rbcL and matK phylogenies, all the purple and green yellow morphs grouped together. For this region, the color morphs also grouped within the same clade as the G. paniculata reference sequence. While G. scorzonerifolia and G. acutifolia have also been recorded in the Great Lakes regions (Pringle, 1976), and have a similar general phenotype as G. paniculata, these species were clearly within a distinct clade that was separate from the two color morphs. Similarly, while G. perfoliata has been reported to be naturalized in North America (Pringle, 1976), it grouped outside of the G. paniculata and color morph cluster.

Our greenhouse germination study showed that seeds collected from mature green-yellow morphs produced a significantly higher proportion of green-yellow individuals than seeds collected from mature purple morphs. Of the seeds collected from mature green-yellow morphs 11% resulted in green-yellow morphs, while only 1% of seeds from mature purple morphs resulted in green-yellow morphs. However, seeds from both color morphs primarily produced purple morphs. The mechanism driving the color difference between the purple and green-morphs is currently unknown. Within SBDNL, the purple morph is the most common form, with green-yellow individuals found interspersed in a couple of locations throughout the dunes (E. Rice, H. Leimbach-Maus, C.G. Partridge, 2017, personal observation). The largest observed group of green-yellow morphs consists of a few hundred plants clumped within approximately an acre-sized area and interspersed throughout large groups of purple morphs. Based upon the dispersal patterns of the two morphs throughout the dunes, and our germination results, the color difference observed does not appear to be solely environmentally driven, and likely has a genetic component. Potential candidate genes that could be influencing these color differences include those involved in the anthocyanin pathway, which influences red-purple coloration in a number of plants (Asen, Stewart & Norris, 1972; De Pascual-Teresa, Santos-Buelga & Rivas-Gonzalo, 2002; Abdel-Aal, Young & Rabalski, 2006). In addition, anthocyanin can rapidly accumulate in the shoots of plants following cold exposure (Leng et al., 2000), and we have observed an increase in the amount of purple coloration in the purple morphs as temperatures decrease (C.G. Partridge, 2017, personal observation). Further work will begin to elucidate the specific mechanism influencing this color difference in invasive G. paniculata populations, as well as to explore whether this color variation drives functional differences between the morphs.

Taken together, these data show that the purple and green-yellow morphs within SBDNL are the same species, and that species is G. paniculata. One concern with the green-yellow morph initially noted by TNC removal crews was that the taproot tended to be more diffuse than the purple morph, potentially making manual removal of these plants less effective. However, we have not noted differences in the taproot structure between these two morphs when grown under controlled conditions (Fig. 7). Additionally, our lab’s observations in the field (C.G. Partridge, H. Leimbach-Maus, 2017, personal observation) have not found any indication that large differences in root structure occur between mature plants of the two color morphs. Therefore, current management approaches for these populations should be maintained to control the further spread of G. paniculata throughout the Michigan coastal dune system.

Figure 7 (A) Green-yellow morph, and (B) common purple morph after 5-months in the GVSU greenhouse.

Note the similarity in taproot structure between the two plants.

Conclusions

Our data show that both the purple and green-yellow color morphs of baby’s breath in Sleeping Bear Dunes National Lakeshore are G. paniculata and the observed color differences likely have some genetic basis. Based on this current information, we recommend that these color morphs continue to be managed in a similar manner and that distinct management strategies do not need to be established at this time.

Supplemental Information

Supplemental Information 1 Files containing raw sequences, compiled sequence files, final sequence alignments, and nexus files.

Click here for additional data file.

Supplemental Information 2 GenBank assession numbers or Barcode of Life Database (BOLD) identifiers for Gypsophila sp. reference sequences.

Click here for additional data file.

Supplemental Information 3 Neighbor-Joining analysis of baby’s breath color morphs in relation to other Gypsophila species for rbcL, matK, and ITS2 combined.

The evolutionary history was inferred using the Neighbor-Joining method (Saitou and Nei 1987). The optimal tree with the sum of branch length = 0.04111371 is shown. The tree is drawn to scale, with branch lengths in the same units as those of the evolutionary distances used to infer the phylogenetic tree. The evolutionary distances were computed using the Tamura 3-parameter method (Tamura, 1992) and are in the units of the number of base substitutions per site. All positions containing gaps and missing data were eliminated. The rate variation among sites was modeled with a gamma distribution.

Click here for additional data file.

Supplemental Information 4 Maximum Parsimony analysis of baby’s breath color morphs in relation to other Gypsophila species for rbcL, matK, and ITS2 combined.

The evolutionary history was inferred using the Maximum Parsimony method. Tree #1 out of 10 most parsimonious trees (length = 55) is shown. The consistency index is (0.930233), the retention index is (0.950820), and the composite index is 0.898957 (0.884483) for all sites and parsimony-informative sites (in parentheses). The percentage of replicate trees in which the associated taxa clustered together in the bootstrap test (500 replicates) are shown next to the branches (Felsenstein 1985). The MP tree was obtained using the Subtree-Pruning-Regrafting (SPR) algorithm with search level 1 in which the initial trees were obtained by the random addition of sequences (10 replicates). The tree is drawn to scale, with branch lengths calculated using the average pathway method and are in the units of the number of changes over the whole sequence. All positions containing gaps and missing data were eliminated.

Click here for additional data file.

Supplemental Information 5 Neighbor-Joining analysis of baby’s breath color morphs in relation to other Gypsophila species for rbcL.

The evolutionary history was inferred using the Neighbor-Joining method (Saitou and Nei 1987). The optimal tree with the sum of branch length = 0.0288485 is shown. The tree is drawn to scale, with branch lengths in the same units as those of the evolutionary distances used to infer the phylogenetic tree. The evolutionary distances were computed using the Jukes-Cantor method (Jukes and Cantor 1969) and are in the units of the number of base substitutions per site. All positions containing gaps and missing data were eliminated.

Click here for additional data file.

Supplemental Information 6 Maximum Parsimony analysis of baby’s breath color morphs in relation to other Gypsophila species for rbcL.

The evolutionary history was inferred using the Maximum Parsimony method. Tree #1 out of 10 most parsimonious trees (length = 12) is shown. The consistency index is (1.000000), the retention index is (1.000000), and the composite index is 1.000000 (1.000000) for all sites and parsimony-informative sites (in parentheses). The percentage of replicate trees in which the associated taxa clustered together in the bootstrap test (500 replicates) are shown next to the branches (Felsenstein 1985). The MP tree was obtained using the Subtree-Pruning-Regrafting (SPR) algorithm (Nei and Kumar 2000) with search level 1 in which the initial trees were obtained by the random addition of sequences (10 replicates). The tree is drawn to scale, with branch lengths calculated using the average pathway method (Nei and Kumar 2000) and are in the units of the number of changes over the whole sequence. All positions containing gaps and missing data were eliminated.

Click here for additional data file.

Supplemental Information 7 Neighbor-Joining analysis of baby’s breath color morphs in relation to other Gypsophila species for matK.

The evolutionary history was inferred using the Neighbor-Joining method (Saitou an Nei 1987). The optimal tree with the sum of branch length = 0.0976890 is shown. The tree is drawn to scale, with branch lengths in the same units as those of the evolutionary distances used to infer the phylogenetic tree. The evolutionary distances were computed using the Tamura 3-parameter method (Tamura, 1992) and are in the units of the number of base substitutions per site. All positions containing gaps and missing data were eliminated.

Click here for additional data file.

Supplemental Information 8 Maximum Parsimony analysis of baby’s breath color morphs in relation to other Gypsophila species for matK.

The evolutionary history was inferred using the Maximum Parsimony method. Tree #1 out of 10 most parsimonious trees is shown (length = 64). The consistency index is (1.000000), the retention index is (1.000000), and the composite index is 1.000000 (1.000000) for all sites and parsimony-informative sites (in parentheses). The percentage of replicate trees in which the associated taxa clustered together in the bootstrap test (500 replicates) are shown next to the branches (Felsenstein 1985). The MP tree was obtained using the Subtree-Pruning-Regrafting (SPR) algorithm (Nei and Kumar 2000) with search level 1 in which the initial trees were obtained by the random addition of sequences (10 replicates). The tree is drawn to scale, with branch lengths calculated using the average pathway method (Nei and Kumar 2000) and are in the units of the number of changes over the whole sequence. All positions containing gaps and missing data were eliminated.

Click here for additional data file.

Supplemental Information 9 Neighbor-Joining analysis of baby’s breath color morphs in relation to other Gypsophila species for ITS2.

The evolutionary history was inferred using the Neighbor-Joining method (Saitou and Nei 1987). The optimal tree with the sum of branch length = 0.20377523 is shown. The tree is drawn to scale, with branch lengths in the same units as those of the evolutionary distances used to infer the phylogenetic tree. The evolutionary distances were computed using the Jukes-Cantor method (Jukes and Cantor 1969) and are in the units of the number of base substitutions per site. All positions containing gaps and missing data were eliminated.

Click here for additional data file.

Supplemental Information 10 Maximum Parsimony analysis of baby’s breath color morphs in relation to other Gypsophila species for ITS2.

The evolutionary history was inferred using the Maximum Parsimony method. Tree #1 out of 10 most parsimonious trees is shown (length = 36). The consistency index is (0.969697), the retention index is (0.993289), and the composite index is 0.965697 (0.963189) for all sites and parsimony-informative sites (in parentheses). The percentage of replicate trees in which the associated taxa clustered together in the bootstrap test (500 replicates) are shown next to the branches (Felsenstein 1985). The MP tree was obtained using the Subtree-Pruning-Regrafting (SPR) algorithm (Nei and Kumar 2000) with search level 1 in which the initial trees were obtained by the random addition of sequences (10 replicates). The tree is drawn to scale, with branch lengths calculated using the average pathway method (Nei and Kumar 2000) and are in the units of the number of changes over the whole sequence. All positions containing gaps and missing data were eliminated.

Click here for additional data file.

We would like to thank Shaun Howard from The Nature Conservancy for his help in identifying the green-yellow morph and Benjamin Giffin for his help with sequencing analysis. We would also like to thank Kurt Thompson and Doug Haywick for assisting with the figure construction, and Alexis Hoskins and Doug Haywick for helping with planting.

Additional Information and Declarations

Competing Interests

Author Contributions

Field Study Permissions

DNA Deposition

Data Availability

The authors declare that they have no competing interests.

Marisa L. Yang conceived and designed the experiments, performed the experiments, analyzed the data, contributed reagents/materials/analysis tools, prepared figures and/or tables, approved the final draft.

Emma Rice conceived and designed the experiments, contributed reagents/materials/analysis tools, approved the final draft.

Hailee Leimbach-Maus conceived and designed the experiments, contributed reagents/materials/analysis tools, approved the final draft.

Charlyn G. Partridge conceived and designed the experiments, performed the experiments, analyzed the data, contributed reagents/materials/analysis tools, prepared figures and/or tables, authored or reviewed drafts of the paper, approved the final draft.

The following information was supplied relating to field study approvals (i.e., approving body and any reference numbers):

Field experiments were approved by the National Parks Service (permit ID SLBE-2015-SCI-0013).

The following information was supplied regarding the deposition of DNA sequences:

The ITS2 sequences are available at GenBank: MG385003–MG385031.

The MatK sequences are available at GenBank: MG603322–MG603346.

The RbcL sequences are available at GenBank: MG547346–MG547373.

The following information was supplied regarding data availability:

The raw sequences, FASTA files, MEGA alignment files, and the nexus file for haplotype network construction are available at Dryad Digital Database: Data from: Identification and characterization of Gypsophila paniculata color morphs in Sleeping Bear Dunes National Lakeshore, MI, USA. Dryad Digital Repository. DOI 10.5061/dryad.j022f80.

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
