# Peer review of "Identification and characterization of Gypsophila paniculata color morphs in Sleeping Bear Dunes National Lakeshore, MI, USA"

_PeerJ, doi:10.7717/peerj.7100_

## Round 0.1 · original submission · Major Revisions

Both reviewers raise major concerns about this work and whether the methodology is appropriate to the question. I would suggest you consider carefully whether a change in emphasis might improve the paper. There are substantial data here that would be interesting to report.

Reviewer 1 ·

Basic reporting

The author’s report on identifying morphological variation found in invasive Gypsophila paniculata populations within the Michigan dune system. I commend the authors for clearly detailing the importance of the question and how different management systems are in use for the different colour morphs. The article is well written, however you should consider expanding the introduction with a brief overview of the taxonomy of Gypsophila and a discussion of the suitability of DNA barcoding to answer your question. The taxonomic overview could highlight why the green-yellow morph could only belong to Gypsophila paniculata and whether other members of the genus are invasive in different ecosystems.

Experimental design

DNA barcoding is well-suited to answer questions like this, however it does not work uniformly well across different plant genera. I believe it would be necessary to show that the chosen plant barcoding regions have the required resolving power to distinguish between a wider range of Gypsophila species. This is even more needed as you mention in lines 76-77 mention that other species of the genus are sold as part of wildflower seed mixes in the region, however you only include G. elegans in your analysis. GenBank and BOLD should be suitable to expand the sampling with a few more species.
The data analysis shows clear results, but you should consider using other barcoding analysis methods, apart from phylogenetic trees, that could further highlight the similarities between the colour morphs and show the difference to any other species (e.g. barcoding gap).

Validity of the findings

Gypsophila paniculata data included in the analysis is suitable to answer the question. However the results would be much more robust if you showed greater context of molecular variation within the genus.

Additional comments

The abstract reads somewhat confusing mentioning two colour morphs for G. paniculata in line 16 but refer to them as potentially different species in line 21. I think it would be clearer to frame the study question similarly as lines 80-81. (Are they the same species or not, rather than the other way round.)
The chloroplast DNA regions should be written as: rbcL, matK throughout the text.
Line 25 sequences were aligned to the reference instead of with the reference.
Line 79 traditional barcode genes needs a citation.
Line 90 Kit instead of Kits.
Line 96 Region instead of gene would be more suitable as ITS2 is not a gene.
Line 100 The citation for the rbcL primers should be the original one: Fay et al. 1997, but do check the primer sequences are the same.
Line 114 Did you mean with forward and reverse primers?
Line 152 As the G. paniculata reference is 16bp away from the samples, the results only suggests similarity between the colour morphs. Additional Gypsophila samples could show a clearer picture.
Line 179 matK is highly variable not conserved. The chloroplast DNA regions in any case most likely would fail at distinguishing cultivars, unless they originate from different maternal plants.
Line 189 may have derived instead of may be derived
Line 193 Whose personal observation was this?
Figure 2 Phylogenetic analysis instead of construction?
Figure 3 The size of the rectangles and the ovals are not readily comparable. It would help if they were all the same type. The figure should have a legend for the different sizes. As lines between the nodes are not dependent on length you could shrink the lines leading to the G. paniculata reference to get a more compact figure, while showing the same relationship.

Reviewer 2 ·

Basic reporting

see below

Experimental design

see below

Validity of the findings

see below

Additional comments

This article is written in a clear language and the methodology used for DNA analysis is updated and correct. References are updated and pertinent.
Anyway I'm quite skeptical about suggesting to accept this manuscript for publication in its present form and with its actual finalities. Infact, it sounds strange to my ears, to listen that the author decided to test if the two taxa belong to the same species, with DNA analysis, in order to plan management activities.
The two taxa (of the same or of different species) occupy the same habitat but have different habit and bahaviour. So, as reported by the same author, the manual procedures that have to be applied by the operators to eradicate the different plants are different.
So the results obtained by the authors and the conclusions reported are not consistent.
I suggest to the authors to present their research under a different light. The dna analysis can be justified to understand if the two taxa can be distinguished at specific or infraspecific level and to see if them coudl have been arrived from different sources, but not to decide the management activities.
Figure 1 includes only the upper part of the 2 taxa, it should include also the lower part, that is reported to be different.

---

## Round 0.2 · accepted · Accept

Thank you for incorporating the suggestions made during review.

Reviewer 1 ·

Basic reporting

The paper has improved greatly with the revision of the molecular work and the seed experiment. I found the story of the paper straightforward and the text easy to read. I have a few comments below, but I am happy with the paper to be published.

Experimental design

OK

Validity of the findings

OK

Additional comments

I would suggest to include the author for each species at the first mention in the text.

rbcL and matK should be italicized throughout the text. ITS2 is correct as normal text.

Line 152: ITS2 is misspelled

Line 173: I think methods would be more suitable than models.

Line 204: G. elegans is missing the 's'

Line 292 & 300: I found these two sentences to be ambiguous. Do you recommend carrying on with the current management as cutting for the purple morph and glyphosphate for the green or cutting for both as they are the same species and seem to show the same taproot structure?

Figure 2-5: These figures need a line about the bootstrap support e.g. numbers at nodes indicate bootstrap support

Figure 6 : I would replace ovals with circles in the caption. I would recommend including the proportion of the colour morphs in the two red haplotype circles as a pie chart. This could show how much the two haplotypes are shared between the colour morphs. PopART should be able to do this.

Reviewer 2 ·

Basic reporting

The authors have applied the requested improvements. Now, on my opinion, the article is presented under the correct perspective. As in the previous version the language used is clear and unambiguous. Figures and data associated are pertinent.

Experimental design

Now the research is well defined. The Methods used are appropriate and are described with sufficient detail.

Validity of the findings

Conclusions are well stated.

Additional comments

I recommend the acceptance of this MS as it is. No further revisions are required.